Integrative growth physiology and transcriptome profiling of probiotic Limosilactobacillus reuteri KUB-AC5

Jatuponwiphat Theeraphol 1
Namrak Thanawat 2
Nitisinprasert Sunee 2
Nakphaichit Massalin 2 fagimln@ku.ac.th
Vongsangnak Wanwipa 3 4 fsciwpv@ku.ac.th
1 Interdisciplinary Graduate Program in Bioscience, Faculty of Science, Kasetsart University , Bangkok , Thailand
2 Department of Biotechnology, Faculty of Agro-Industry, Kasetsart University , Bangkok , Thailand
3 Department of Zoology, Faculty of Science, Kasetsart University , Bangkok , Thailand
4 Omics Center for Agriculture, Bioresources, Food, and Health, Kasetsart University (OmiKU) , Bangkok , Thailand
Czajkowski Robert
Electronic publication date: 2021 Oct 5
Publication date: 2021
Volume: 9
Electronic Location ID: e12226
Received 2021 Jun 3; Accepted 2021 Sep 8
Copyright: © 2021 Jatuponwiphat et al.
Copyright year: 2021
Copyright holder: Jatuponwiphat et al.
License: This is an open access article distributed under the terms of the Creative Commons Attribution License, which permits unrestricted use, distribution, reproduction and adaptation in any medium and for any purpose provided that it is properly attributed. For attribution, the original author(s), title, publication source (PeerJ) and either DOI or URL of the article must be cited.
License URL: https://creativecommons.org/licenses/by/4.0/

Keywords: Limosilactobacillus reuteri KUB-AC5, Growth physiology, Inhibition effects, Transcriptome profiling

Funding: Department of Zoology, Faculty of Science and Kasetsart University Research and Development Institute (KURDI) at Kasetsart University Science Achievement Scholarship of Thailand (SAST) Interdisciplinary Graduate Program in Bioscience, Faculty of Science, Kasetsart University The authors received funding from the Department of Zoology, Faculty of Science and Kasetsart University Research and Development Institute (KURDI) at Kasetsart University; the Science Achievement Scholarship of Thailand (SAST); and the Interdisciplinary Graduate Program in Bioscience, Faculty of Science, Kasetsart University. The funders had no role in study design, data collection and analysis, decision to publish, or preparation of the manuscript.

==============================
Limosilactobacillus reuteri KUB-AC5 has been widely used as probiotic in chicken for Salmonella reduction. However, a preferable carbon source and growth phase is poorly characterized underlying metabolic responses on growth and inhibition effects of L. reuteri KUB-AC5. This study therefore aimed to investigate transcriptome profiling of L. reuteri KUB-AC5 revealing global metabolic responses when alteration of carbon sources and growth phases. Interestingly, L. reuteri KUB-AC5 grown under sucrose culture showed to be the best for fast growth and inhibition effects against Salmonella Enteritidis S003 growth. Towards the transcriptome profiling and reporter proteins/metabolites analysis, the results showed that amino acid transport via ABC systems as well as sucrose metabolism and transport are key metabolic responses at Logarithmic (L)-phase of L. reuteri KUB-AC5 growth. Considering the Stationary (S)-phase, we found the potential reporter proteins/metabolites involved in carbohydrate metabolism e.g., levansucrase and levan. Promisingly, levansucrase and levan were revealed to be candidates in relation to inhibition effects of L. reuteri KUB-AC5. Throughout this study, L. reuteri KUB-AC5 had a metabolic control in acclimatization to sucrose and energy pools through transcriptional co-regulation, which supported the cell growth and inhibition potentials. This study offers a perspective in optimizing fermentation condition through either genetic or physiological approaches for enhancing probiotic L. reuteri KUB-AC5 properties.

Introduction

Probiotics are defined as microorganisms that can live in the gastrointestinal tract and confer benefits to the host health (FAO/WHO, 2001). Several mechanisms of probiotics on health have been reported. Probiotics have an essential role in activating immune-competent cells in the intestine (Corthésy, Gaskins & Mercenier, 2007; Kemgang et al., 2014). They play an important role in pathogen elimination through their antimicrobial activity and adhesion properties to gut epithelial mucosa and colonization in the microbial ecosystem (Fonseca et al., 2021). The use of probiotics in livestock feeds has increased considerably in the last decade. The majority of the currently used probiotics are based on lactic acid bacteria, mainly Enterococcus spp. and Lactobacillus spp., which are used in functional foods (Haghshenas et al., 2017; Shi et al., 2016). Among well-known probiotics, Limosilactobacillus reuteri KUB-AC5 isolated from the chicken intestine displays an effective inhibition activity against a broad spectrum of the growth of gram-positive and gram-negative bacteria including Escherichia coli and Salmonella spp. (Nakphaichit et al., 2011; Nitisinprasert et al., 2000). Nakphaichit et al. (2018) found that the strain KUB-AC5 for 107 CFU/ml inhibited pathogenic bacteria e.g., Salmonella. Besides, it could also promoted the growth of chicken and additionally inhibited Klebsiella, Chryseobacterium, Citrobacter, Aeromonas, and Acinetobacter (Nakphaichit et al., 2011).

Regarding on cultivation process, as known, nutrient components in the medium have a great influence on the growth and biomass (Dewi et al., 2020; Petrut et al., 2019). Here, the carbon source is one of the most important influencing factors for growth, biomass and desirable products. Thus, an optimization of carbon source has been the subject of continuous biotechnological research (Begovic et al., 2010).

As omics technology advances, genomics and transcriptomics have revolutionized biology of probiotic bacteria. Recently, genome sequencing and analysis of L. reuteri KUB-AC5 (Jatuponwiphat et al., 2019) revealed genes involved in cobalamin (vitamin B12) and folate (vitamin B9) biosynthesis, the unique biosynthetic gene clusters (e.g., exopolysaccharide), as well as the bacteriocin-related genes. This allows us to understand probiotic properties of L. reuteri KUB-AC5. Beyond, transcriptomics can indeed identify the RNA transcripts produced by the genome at a given time under specific circumstance that provides a linkage between the genotype and the particular phenotype (Lowe et al., 2017). However, in context of gene expression at a large scale by RNA sequencing and functional identification of L. reuteri KUB-AC5 has not yet been done upon different conditions.

This study therefore aimed to investigate transcriptome profiling of L. reuteri KUB-AC5 revealing global metabolic responses when alteration of carbon sources and growth phases. To obtain informative data, carbon sources and growth phases were subjected as variables for the bacterial cultivation. The comparative transcriptomic analysis of the glucose and sucrose cultures grown under the Logarithmic (L)-phase and Stationary (S)-phase were explored for identifying the differentially expressed genes (DEGs). The metabolic network of L. reuteri KUB-AC5 at a genome-scale was also constructed for integrating with DEGs to analyze the reporter proteins/metabolites associated with growth and inhibition effects. This study proposes the metabolic responses of L. reuteri KUB-AC5 on the preferable carbon source and growth phase by cooperating in several pathways through the transcriptional expression of a set of metabolic genes involving growth and inhibition potentials. This offers a perspective in optimizing fermentation condition through either genetic or physiological approaches for enhancing probiotic L. reuteri KUB-AC5 properties.

Materials and Methods

Culture conditions of L. reuteri KUB-AC5

L. reuteri KUB-AC5 isolated from chicken intestine samples was obtained from the stock culture of Department of Biotechnology, Faculty of Agro-industry, Kasetsart University, Thailand (Nitisinprasert et al., 2000). This strain was preserved at –80 °C in MRS medium (Difco) containing 20% (v/v) glycerol. Culture was propagated twice in MRS medium at initial pH 6.5 and 37 °C for 18–24 h. All cultivations started with the approximate concentration of 109 CFU/ml at late logarithmic phase of L. reuteri as the inoculum for further investigating inhibition effects (Sobanbua et al., 2020). It was cultured with two independent replicates at 37 °C for 24 h in a modified MRS medium (Hamsupo et al., 2005) with either 2% (w/v) sucrose or 2% (w/v) glucose. Samples were taken at different time points (i.e., 0, 3, 6, 9, 12, 15, 18, 21 and 24 h) for further viable cell count and pH measurement. The viability of L. reuteri KUB-AC5 was determined by the standard plate count method. Aliquots of one mL of each sample were diluted by nine mL of sterile 0.85% NaCl solution in serial decimal dilutions. It was afterwards plated in MRS agar (Himedia) and incubated under anaerobic conditions at 37 °C for 48 h. The cell count was performed as log of colony forming units per milliliter (log CFU/ml). For pH measurement, it was performed by a digital pH meter (Mettler Toledo S220).

Determination of inhibition effects of L. reuteri KUB-AC5

Inhibition effects of the L. reuteri KUB-AC5 was determined by 96-well microtiter plate assay according to the method of Sobanbua et al. (2020). In brief, Salmonella Enteritidis S003 grown in nutrient broth (NB) for 12 h used as an indicator strain. The cell pellet obtained from 100 μl of the diluted culture was suspended in two-fold concentrated NB at pH five with approximate concentration of 105 CFU/ml. Then, 100 μl of supernatant of samples was transferred to each of microtiter plate and mixed well by pipetting. Growth of the indicator strain at 37 °C for 24 h was determined by optical density (OD) at 600 nm using a Bio-Rad microtiter reader model 450 (Bio-Rad Laboratories, Hercules, CA, USA). Notably, growth of S. Enteritidis S003 in NB at pH five was also used as a reference for investigating inhibition effects (Tangthong & Nitisinprasert, 2012).

RNA extraction towards sequencing of L. reuteri KUB-AC5

For RNA extraction, cells were initially harvested from two independent cultures grown at two different carbon sources and growth phases i.e., glucose or sucrose and L-phase or S-phase. It was noted that harvested cells were taken at 6 h (mid L-phase) for investigating expression of growth-related genes. Besides, harvested cells were also taken at 12 h (early S-phase) as initial transition of growth ceases for further capturing expression of genes associated with inhibition effects against S. Enteritidis S003 growth. After that the harvested cells were quickly frozen in liquid nitrogen, and then stored at −80 °C. Total RNA samples were then extracted using the TRIzol reagent (Invitrogen, Santa Clara, CA, USA) according to the manufacturer’s recommendations. For purity determination, both contaminants and degradation components of RNA samples were evaluated using ND-1,000 spectrophotometer (NanoDrop Technologies Inc., Waltham, MA, USA) and 1% agarose gel electrophoresis, respectively. RNA integrity number (RIN) (Schroeder et al., 2006) was assessed using Agilent 2,100 Bioanalyzer (Agilent Technologies, Santa Clara, CA, USA). In regard to RNA concentration, the Qubit® RNA assay kit was used in a Qubit® 2.0 Fluorometer (Life Technologies, Carlsbad, CA, USA). The purified RNA samples were then used for cDNA library construction (Novogene Bioinformatics Technology Co. Ltd., Beijing, China) and RNA sequencing was finally performed by Illumina NovaSeq 6,000.

Transcriptome data analysis and functional annotation of L. reuteri KUB-AC5

To analyze transcriptome data, initially raw reads in FASTQ format were first processed by removing all possible low quality reads e.g., adaptor, unknown nucleotides (N) > 10%, and Q20 < 50%. The clean reads from all studied conditions (glucose or sucrose at L-phase or S-phase) were then mapped to the L. reuteri KUB-AC5 genome (Jatuponwiphat et al., 2019) using the alignment package Burrows–Wheeler Alignment tool (BWA) (Li & Durbin, 2009). The resulting mapping files in SAM format were afterwards converted to BAM format using SAMtools version 1.6 (Li et al., 2009) for estimating mapped reads to genes reference. Gene expression levels were estimated using FPKM (Fragments Per Kilobase per Million mapped reads) values based on RNA-Seq data (Trapnell et al., 2012). Once considering FPKM value ≥ 1, gene functions were consequently annotated based on the following integrative databases: the non-redundant (NR) protein database, the SWISS-PROT protein database (Bairoch & Apweiler, 2000), the database of Clusters of Orthologous Groups of proteins (COGs) (Tatusov et al., 2000), and the Kyoto Encyclopedia of Genes and Genomes (KEGG) database (Kanehisa et al., 2010).

Further, DEGs analysis was performed between possible pairwise comparisons between glucose and sucrose at L-phase or S-phase of L. reuteri KUB-AC5 cultures using DESeq2 package (Anders & Huber, 2010; Love, Huber & Anders, 2014). DEGs between two possible conditions were calculated based on the relative ratios of the FPKM values under threshold of FPKM value ≥ 1 and |log2 fold change| ≥ 1.5. In addition, the false discovery rate (FDR) was used to adjust p-value for the significance of the differences. In this study, the FDR < 0.05 was set for gaining significant genes (Marioni et al., 2008).

Identification of reporter proteins/metabolites in relation to growth and inhibition effects of L. reuteri KUB-AC5

To identify the reporter proteins/metabolites, the metabolic network of L. reuteri KUB-AC5 at a genome-scale was constructed by annotated data (Jatuponwiphat et al., 2019) and KEGG pathway mapping (Kanehisa et al., 2016). Then, the constructed metabolic network of L. reuteri KUB-AC5 as a scaffold was integrated with the list of DEGs in response to preferable carbon source and growth phase using R package Piano (Platform for Integrated Analysis of Omics data) (Väremo, Nielsen & Nookaew, 2013). Notably, the protein or metabolite, which had a distinct up-directional P-value < 0.05, was identified as a reporter designating protein or metabolite around which the most significant transcriptional changes occurred (Patil & Nielsen, 2005; Väremo, Nielsen & Nookaew, 2013). Overlaying reporter protein/metabolite on phenotype data e.g., cellular growth and inhibition effects of L. reuteri KUB-AC5, the global metabolic responses of L. reuteri KUB-AC5 would be uncovered.

Results

Growth characteristics and inhibition effects of L. reuteri KUB-AC5 on different carbon sources and growth phases

The growth characteristics of L. reuteri KUB-AC5 on different carbon sources and growth phases are shown in Fig. 1A. Observably, L. reuteri KUB-AC5 was rapidly growth (>7 log CFU/ml) during L-phase (3–6 h) of both sucrose and glucose cultures (Table S1). The maximum specific growth rate (μmax) of 0.05 h−1 was obtained in the sucrose culture, which was significantly higher than that of the glucose culture (μmax of 0.04 ± 0.001 h−1), as well as the bacterial biomass productivity was significantly higher in the sucrose culture (0.36 log CFU/ml/h) than in the glucose culture (0.28 log CFU/ml/h) under P-value < 0.05 (Table S2). As observed in Table S3, the similar pH values were noted in both sucrose and glucose cultures across different time points. As a result, this suggests that the sucrose was preferable source of carbon for L. reuteri KUB-AC5 growth.

Figure 1 Growth physiology of L. reuteri KUB-AC5 under different carbon sources and growth phases.

(A) Viable cell count (B) Cell-free supernatant (CFS) of L. reuteri KUB-AC5 on inhibition effects against Salmonella Enteritidis S003 growth. Note: Statistical significance under P-value < 0.05 was considered for investigating growth characteristics (Table S2) and inhibition effects of L. reuteri KUB-AC5 (Table S4).

To further explore inhibition effects of L. reuteri KUB-AC5, cell-free culture supernatants (CFS) of L. reuteri KUB-AC5 from sucrose or glucose cultures at L-and S-phases were grown in NB at pH five for investigating the growth of S. Enteritidis S003. Here, S. Enteritidis S003 growth was determined at different time points (i.e., 0, 5, 7, 10, 12, 16, 18 and 24 h). As the result in Fig. 1B, this indicates that the CFS of L. reuteri KUB-AC5 from sucrose culture at S-phase showed the highest inhibition effects against S. Enteritidis S003 growth, followed by sucrose at L-phase. It is noted that CFS of L. reuteri KUB-AC5 from glucose culture in such S-phase of growth significantly showed less inhibition effects against S. Enteritidis S003 growth (Fig. 1B and Table S4) when compared to sucrose culture (P-value < 0.05).

Assessment and analysis of DEGs data of L. reuteri KUB-AC5

As different culture conditions by various carbon sources and growth phases resulted in altered phenotypes, it was of interest to investigate transcriptional changes of L. reuteri KUB-AC5 on growth and inhibition effects. L. reuteri KUB-AC5 mRNA pools isolated from glucose or sucrose culture at L-phase or S-phase were sequenced using an Illumina NovaSeq 6,000. As a result, raw reads of 25.14 Megabase pairs (Mb) were gained. After removing adaptor and low-quality sequences and read pollution, clean reads were finally retrieved with an average of 24.92 Mb and an average sequencing quality of 98.37% (Table 1, Table S5). Further, a total of clean reads was then mapped to the L. reuteri KUB-AC5 genome (Jatuponwiphat et al., 2019) which resulted in the total mapped reads average for 86.29% (Table 1). Accordingly, this resulted in 1,945 expressed genes (Table S6). Regarding the annotation of 1,945 expressed genes, 1,357 genes (FPKM value ≥ 1) were considered and annotated functions by COGs database. The results show in Fig. 2. Once considering three selected functional categories, the metabolism (671 genes), the information storage and processing (397 genes) and cellular processes and signaling (289 genes) were identified. Observably, the majority of expressed genes of L. reuteri KUB-AC5 were associated with metabolic functions (Fig. 2A). Interestingly, we found the highest number of genes involved in the amino acid transport and metabolism (142 genes), followed by carbohydrate transport and metabolism (117 genes), coenzyme transport and metabolism (95 genes), nucleotide transport and metabolism (88 genes), energy production and conversion (77 genes), inorganic ion transport and metabolism (67 genes), lipid transport and metabolism (60 genes), as well as secondary metabolites biosynthesis, transport and catabolism (25 genes) (Fig. 2B, Table S7).

Figure 2 Categorizing functions of the expressed genes of L. reuteri KUB-AC5 based on COGs classification.

(A) A pie chart shows a distribution of functions into three selected COGs functional categories. (B) A horizontal bar chart shows a distribution of genes across sub-metabolism categories.

Table 1 Mapping results of L. reuteri KUB-AC5 transcriptome.

Summary	Glucose	Sucrose	Average	
L-phase	S-phase	L-phase	S-phase	
Rep. 1	Rep. 2	Rep. 1	Rep. 2	Rep. 1	Rep. 2	Rep. 1	Rep. 2	
Total clean reads (Mb)	26.84	25.35	22.72	28.13	24.89	22.97	26.26	22.19	24.92	
Total mapped reads to genome (Mb)	24.10 (86.7%)	22.85 (86.8%)	19.07 (81.2%)	24.52 (84.2%)	22.94 (89.0%)	20.65 (86.8%)	23.94 (88.1%)	20.07 (87.5%)	22.27 (86.29%)	
Number of expressed genes	1,934	1,934	1,941	1,939	1,940	1,936	1,943	1,936	1,938	
All expressed genes	1,945	

To further analyze DEGs, transcriptome data were organized into two pairwise comparisons: sucrose versus glucose at L-phase or S-phase. Under the thresholds of |log2 (fold change)| ≥ 1.5 with a false discovery rate (FDR) ≤ 0.05 between all possible comparisons (Fig. 3A). Accordingly, the numbers of significant genes from DEGs analysis were identified. Observably, these genes were distributed into sucrose versus glucose conditions at L-phase (214 significant genes) or S-phase (151 significant genes) were identified. Noticeably, we found 52 up-regulated genes and 162 down-regulated genes in sucrose at L-phase as well as 39 up-regulated genes and 112 down-regulated genes in sucrose at S-phase (Fig. 3B). Figure S1 shows an overall list of significant genes from transcriptome profiling.

Figure 3 Differentially expressed genes analysis across pairwise carbon source comparisons.

(A) Volcano plots show the DEGs under −log10 (FDR) against the log2 (fold change) between pairwise carbon source comparisons at L-or S-phases. The red and blue dots represent significantly up-and down-regulated genes (FDR < 0.05), respectively. Gray dots are not significant genes. (B) Horizontal bar chart shows the number of significant genes in each pairwise comparison set.

Reporter proteins/metabolites uncovering global metabolic responses from the integrative transcriptome and metabolic network of L. reuteri KUB-AC5

To identify reporter proteins/metabolites, the constructed metabolic network of L. reuteri KUB-AC5 containing 609 genes, 622 biochemical reactions, and 698 metabolites (Table S8) was used for integrative transcriptome analysis (see “Materials and Methods”). Promisingly, we found 11 reporter proteins under distinct up-directional P-value < 0.05 (Table 2). Of considering these reporter proteins in context of expression patterns, we clearly observed the higher transcriptional changes in a form of log2 FPKM values occurred in sucrose than glucose in both L-and S-phases as illustrated in Figs. 4 and 5. Once considering at each phase i.e., L-phase, we observed unique reporter proteins i.e. L-histidine transport via ABC system, L-arginine transport via ABC system, L-glutamine transport via ABC system, L-cysteine transport via ABC system, and D-cysteine transport via ABC system. Regarding at S-phase, we found unique reporter proteins, namely levansucrase, sucrose phosphorylase, and sucrose transport via permease, and dihydroorotase. Clearly, the results showed that amino acid transport via ABC systems and sucrose metabolism and transport are key metabolic response-related insight from the transcriptome profiling of L. reuteri KUB-AC5.

Figure 4 A heat map diagram shows different gene patterns of reporter proteins across sucrose versus glucose condition at L-phase of growth.

Each gene is colored by log2 FPKM value.

Figure 5 A heat map diagram shows different gene patterns of reporter proteins across sucrose versus glucose condition at S-phase of growth.

Each gene is colored by log2 FPKM value.

Table 2 List of reporter proteins illustrates global metabolic responses when alterations of carbon sources and growth phases of L. reuteri KUB-AC5.

Reporter proteins	Distinct up-directional P-value*	
L-phase of growth		
L-histidine transport via ABC system	0.001996	
L-arginine transport via ABC system	0.001996	
L-glutamine transport via ABC system	0.005988	
Carbamoyl-phosphate synthase (glutamine-hydrolysing)	0.011976	
L-cysteine transport via ABC system	0.011976	
D-cysteine transport via ABC system	0.011976	
S-phase of growth		
Carbamoyl-phosphate synthase (glutamine-hydrolysing)	0.017964	
Levansucrase	0.021956	
Sucrose phosphorylase	0.043912	
Dihydroorotase	0.043912	
Sucrose transport via permease	0.043912	
Note:

* Under distinct up-directional P-value < 0.05.

Considering on the reporter metabolites as identified in Fig. 6, once shifting from glucose to sucrose at L-phase or S-phase, accordingly we found 14 reporter metabolites in common e.g., sucrose, D-fructose, ATP, ADP, phosphate, diphosphate, L-glutamate, L-glutamine, carbamoylphosphate, N-carbamoyl-L-aspartate, dihydroorotate, orotidine 5′-phosphate, orotate, and bicarbonate. These revealed that L. reuteri KUB-AC5 had a metabolic control in acclimatization to carbon sources (e.g., sucrose and fructose) and energy pools (e.g., ATP) through transcriptional co-regulation, which supported the cell growth at L-or S-phases cultivation.

Figure 6 A subnetwork of reporter metabolites across sucrose versus glucose condition.

(A) at L-phase of growth and (B) at S-phase of growth. The node size corresponds to the size of the gene sets and the edge thickness represents the number of shared genes. P-value is taken from distinct up-directional P-value.

Regarding on unique metabolites identified in each phase, interestingly we majorly found different amino acids and their intermediates involving in the metabolism of the amino acids (e.g., cysteine, histidine, aspartate, arginine, L-phenylalanine, L-tyrosine, 4-aminobutanoate, phenylpyruvate, N(omega)-(L-arginino)succinate, 3-(4-hydroxyphenyl)pyruvate), 2-oxoglutarate as reporters at L-phase (Fig. 6A, Table S9). These suggest that amino acid metabolism is a key role for L. reuteri KUB-AC5 primary growth. Considering on S-phase, it was the depletion of an essential nutrient and/or the formation of an inhibitory product. Promisingly, we found a potential unique reporter metabolite e.g., levan (Fig. 6B, Table S10) as an inhibitory product. For the other remaining reporter metabolites in S-phase, we also found 5-phospho-alpha-D-ribose1-diphosphate, D-glucose-1-phosphate, D-ribulose 5-phosphate, formate, 2,5-diamino-6-hydroxy-4-(5′-phosphoribosylamino)-pyrimidine, 3,4-dihydroxy-2-butanone-4-phosphate. Taken together, the reporter proteins/metabolites allow identifying hot spots around which significant regulation occurs in amino acid metabolism and transport as well as sucrose metabolism and transport of L. reuteri KUB-AC5.

Identified key gene and metabolic function in relation to inhibition potentials of L. reuteri KUB-AC5

Focusing reporter proteins/metabolites in relation to phenotype data e.g., the highest inhibition effects of L. reuteri KUB-AC5 against S. Enteritidis S003 growth in sucrose culture at S-phase (Fig. 1B), interestingly we found that significant regulation occurred around levan as listed in Table 3. Indeed, we observed sacB genes in a form of cluster (e.g., AC5u0009GL000236, AC5u0009GL000237, AC5u0009GL000238 and AC5u0009GL000239) with overall up-regulation in sucrose. According to S-phase, observably we found higher log2 fold changes than L-phase (Table 3). Altogether, these clearly indicated that the expression of sacB gene cluster was highly induced by the sucrose at S-phase. Mapping of phenotypic data of inhibition effects and transcriptome profiling therefore suggests that levan biosynthesis from sucrose utilization via levansucrase might be occurred, so that levan can be a target of interest for inhibition potentials of probiotic L. reuteri KUB-AC5.

Table 3 List of up-regulated sacB genes in a cluster encoding levansucrase from L. reuteri KUB-AC5 transcriptome under sucrose culture.

Gene ID	Gene symbol	L-phase of growth	S-phase of growth	Description	
log2FC	Status	log2FC	Status	
AC5u0009GL000236	sacB	0.65	Up	2.92	Up	Levansucrase	
AC5u0009GL000237	1.70	3.35	
AC5u0009GL000238	1.76	2.80	
AC5u0009GL000239	2.43	3.49	

Discussion

The important effects of probiotics against pathogens were inhibition activity by antimicrobial substance production (Prabhurajeshwar & Chandrakanth, 2019). As reported by Iordache et al. (2008), they revealed the presence of bioactive molecules produced by lactic acid bacteria with probiotic potential which the expression of opportunistic bacterial virulence factors could be suppressed. L. reuteri KUB-AC5 isolated from chicken intestine exerted antimicrobial substance exhibiting inhibition activity against pathogen, especially Salmonella spp. (Nakphaichit et al., 2011; Nitisinprasert et al., 2000). From physiological studies for maximized growth characteristics and inhibition effects of L. reuteri KUB-AC5, the results clearly indicated that sucrose is suggested to be an inducer for fast growth and used to increase inhibition potentials of L. reuteri KUB-AC5 (Fig. 1). It is worth to note that other carbon source e.g., glucose might be alternatively used, however less inhibition effects were observed upon activity of antimicrobial substance. Exploring sucrose on different growth phases, significant genes from DEGs analysis together with reporter proteins/metabolites identification revealed that global metabolic responses of L. reuteri KUB-AC5. At L-phase of growth, the reporter proteins/metabolites involving in the metabolism of the amino acids transport via ABC systems. Generally, nutrient uptake by amino acids transporters via ABC system is important for bacterial survival, such as L-glutamate as reporter metabolite (Fig. 6A), an essential amino acid for bacterial growth and intermediate product for ammonium assimilation. L-glutamate is thus a precursor for the synthesis of the antioxidant glutathione (Amon, Titgemeyer & Burkovski, 2010; van Heeswijk, Westerhoff & Boogerd, 2013). This suggests that amino acids are essential nutrients that lead to stimulate or promote the growth of L. reuteri KUB-AC5. Considering on S-phase, we found potential reporter protein, namely levansucrase encoding by sacB gene (Table 3). The most interesting feature of levansucrase was its distinctive ability to produce levan from sucrose (Srikanth et al., 2015).

Generally, as known sacB gene is encoded for the levansucrase (EC: 2.4.1.10), which is activated in the presence of sucrose and secreted out from the cell by the SecA pathway into the culture medium (Dedonder, 1966; Marvasi, Visscher & Casillas Martinez, 2010; Pereira et al., 2001; Tanaka et al., 1981). As earlier reports, levansucrase was involved in sucrose hydrolysis and levan biosynthesis (Jäger et al., 1992; Leloup et al., 1999; Limoli, Jones & Wozniak, 2015; Osman, Fett & Fishman, 1986) in L. reuteri strains 121, 100–23 and LTH5448 (Ni et al., 2018; Ozimek et al., 2006; Sims et al., 2011; van Hijum et al., 2004; van Hijum et al., 2001). In context of pharmaceutical functions, levan has been reported that it exhibits antibacterial (Byun, Lee & Mah, 2014), antiviral (Esawy et al., 2011), anti-obesity (Oh et al., 2014), anti-diabetes (Dahech et al., 2011), anti-oxidant (Dahech et al., 2013), and hypolipidemic activities (Belghith et al., 2012), calcium absorption (Kim et al., 2004), and immunostimulation (Xu et al., 2006). Taken together with the potential reporter metabolite, levan (Fig. 6B) showed to be the unique one associated with S-phase in sucrose culture. Due to levan is a type of functional polysaccharide which might increase inhibition effects against indicator strain S. Enteritidis S003 growth (Fig. 1B) in sucrose culture as in agreement on the effect of sucrose induced levan production of L. reuteri strain LTH5448 (Ni et al., 2018). Therefore, these altogether suggest that L. reuteri KUB-AC5 might produce a levansucrase, which synthesizes levan from sucrose.

In aspects of inhibition potentials of L. reuteri KUB-AC5 throughout this study, it was not only levan as a target of interest, but bacteriocins and antimicrobial peptides were also considered. Generally, L. reuteri KUB-AC5 can produce bioactive peptides known as bacteriocins that possess antimicrobial activity against pathogenic bacteria (Sobanbua et al., 2019). Of considering the DEGs of bacteriocins associated genes in L. reuteri KUB-AC5 (e.g., immunity, regulator and transporter) (Jatuponwiphat et al., 2019) in responses to alteration of carbon sources or growth phases, however there was no significant difference (Table S11). Even though, Sobanbua et al. (2020) recently reported that antimicrobial peptide-associated genes were existed in L. reuteri KUB-AC5. Nonetheless, our transcriptome studies suggest that no significant differences were found when alteration of carbon sources or growth phases (Table S12). Taken together, these might suggest that bacteriocins and antimicrobial peptides were produced in specific fermentation conditions (Kaškonienė et al., 2017; Soltani et al., 2020).

Regarding on potential probiotic properties against S. Enteritidis growth by integrative physiology studies, transcriptome profiling, and metabolic network of L. reuteri KUB-AC5 towards reporter proteins/metabolites analysis, these clearly suggest that levan might be a target of interest for inhibition potentials of probiotic L. reuteri KUB-AC5.

Conclusions

Throughout this study, L. reuteri KUB-AC5 had a metabolic control in acclimatization to preferable carbon sources (e.g., sucrose) and energy pools through transcriptional co-regulation, which supported the cell growth and inhibition potentials.

Availability of data and materials

All sequences have been deposited in the Sequence Read Archive (https://www.ncbi.nlm.nih.gov/sra) under the accession number PRJNA563573 (BioSamples: SAMN18437211, SAMN18437246, SAMN18437399, SAMN18437752, SAMN18438585, SAMN18438691, SAMN18439203 and SAMN18439842).

Supplemental Information

Supplemental Information 1 Supplemental Tables S1–S12.

Click here for additional data file.

Supplemental Information 2 A heat map diagram shows overall list of significant genes from transcriptome profiling across sucrose versus glucose condition.

(A) at L-phase of growth and (B) S-phase of growth. Each gene is colored by log2 FPKM value.

Click here for additional data file.

The authors would like to thank the Department of Zoology, Faculty of Science, Department of Biotechnology, Faculty of Agro-Industry, Kasetsart University for support, as well as the SciKU Biodata Server, Faculty of Science, Kasetsart University for computing facilities. The authors would like to thank the SciKU Biodata Server, Faculty of Science, Kasetsart University for computing facilities, and the International SciKU Branding (ISB) and the Faculty of Science, Omics Center for Agriculture, Bioresources, Food, and Health, Kasetsart University (OmiKU) for providing resources. W.V. would like to thank the Department of Zoology, International SciKU Branding (ISB), Faculty of Science, Omics Center for Agriculture, Bioresources, Food, and Health, Kasetsart University (OmiKU) for support.

Additional Information and Declarations

Competing Interests

Author Contributions

DNA Deposition

Data Availability

The authors declare that they have no competing interests.

Theeraphol Jatuponwiphat performed the experiments, analyzed the data, prepared figures and/or tables, authored or reviewed drafts of the paper, and approved the final draft.

Thanawat Namrak performed the experiments, prepared figures and/or tables, authored or reviewed drafts of the paper, and approved the final draft.

Sunee Nitisinprasert analyzed the data, authored or reviewed drafts of the paper, and approved the final draft.

Massalin Nakphaichit conceived and designed the experiments, authored or reviewed drafts of the paper, and approved the final draft.

Wanwipa Vongsangnak conceived and designed the experiments, analyzed the data, authored or reviewed drafts of the paper, and approved the final draft.

The following information was supplied regarding the deposition of DNA sequences:

All sequences are available in the Sequence Read Archive: PRJNA563573.

The following information was supplied regarding data availability:

The raw data is available in Supplementary Files.

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
