# Peer review of "Integrative growth physiology and transcriptome profiling of probiotic Limosilactobacillus reuteri KUB-AC5"

_PeerJ, doi:10.7717/peerj.12226_

## Round 0.1 · original submission · Major Revisions

Please address the concerns of both reviewers.

Reviewer 1 ·

Basic reporting

1. For the “abstract” part, too many unnecessary details are mentioned. Mainly findings and conclusions should be listed here other than detailed experimental method and the process;
2. The counting unit should be unified of the whole paper, such as line 49-51, both for the description of KUB-AC5, CFU/g and CFU/ml are listed.
3.The language writing of the whole paper should be better polished.

Experimental design

Some explanations are missed which should be more cleared. eg. In line 88, why you start with the concentration at about 109 CFU/ml? Any supported reference or past experience?

Validity of the findings

Some details of the findings should explain more clearly.eg. In fig.1B, as mentioned, sucrose-based culture showed higher inhibition effect than glucose-based culture;
It seems there is a big turnover in the curve of “glucose at L-phase” at the time of 18-24h, is there any reason for this change? And, if the experiments last for a longer time, will the curve continue to go down? How about the other three?

Additional comments

The paper investigated how the transcriptome change after they found the phenotypes change by cultured with various carbon source. Further, they explored metabolic network and three main functional gene categories which were believed to be the reason of this change.
It is an interesting story standing on the inhibition effect difference of KUB-AC5. Still, here are some questions and comments:
1. Based on the findings of different phenotypes, an investigation of transcriptional change is announced. However, this logical progression is far-fetched. The change of transcriptome is a complex process which determined by various facts.
2. Three functional DEG categories were tested to explain the how the transcriptome changes and how it gives an impact on the phenotypes. Besides these three, have you considered any other macroscopic may also influence its gene expression? Are these three sites enough to result in this change?
3. There are too much quoted contents show in the “results discussion” part which is not common. They are supposed to be described in the “introduction” or “experiments design” parts.
4. Too much repeat description if “results” and “discussion”, the paper is lack of further discussion of how the phenotypes change results in transcriptome change and how is their feedback regulation between them.
5. Too much keywords listed, some of them are meaningless.

Reviewer 2 ·

Basic reporting

See below

Experimental design

See below

Validity of the findings

See below

Additional comments

This article titled “Integrative growth physiology and transcriptome profiling of probiotic Limosilactobacillus reuteri KUB-AC5” reported the growth phases of KUB-AC5 strain in medium with two different carbon sources. The authors carried out a transcriptome analysis combined with growth physiology to predict global metabolic responses using metabolic network at a genome-scale of this strain. The subject is interesting and the data mining is proper. It can be of significance for the use of bioinformatics and omics to analyze the potential beneficial activity of probiotics. However, in this manuscript, I think there are still some problems about the inhibition effects of this strain to be solved.

1. The authors says “Samples were taken at different time points (i.e. 0, 3, 6, 9, 12, 15, 18, 21 and 24 h) for further viable cell count and pH measurement.” (line 91-92). But the pH changing with time is not mentioned in the result. This is very important, because the decrease of pH will also inhibit the growth of some pathogens.

2. The authors took the harvested cells to perform RNA-seq at 6h for L-phase and 12h for S-phase (line 113). Please explain why choose 12h for S-phase? The inhibition effects against S. Enteritidis S003 showed the largest difference appeared at 18h between Sucrose phase-S and Glucose phase-S or between Sucrose phase-L and Glucose phase-L (Fig.1B), while the differences between groups at 12h were less. Samples at this time point may result in a smaller set of differentially expressed genes, and may miss out on important candidate genes associated with the effect of S. Enteritidis S003 inhibition.

3. There is no statistical test in the results of KUB-AC5 strain growth characteristics and S. Enteritidis S003 inhibition (line 164-182). The corresponding P-value should be supplemented in the paragraphs and in Fig 1.

4. The authors used “cell-free culture supernatants (CFS) of L. reuteri KUB-AC5 from sucrose or glucose cultures at L- and S-phases were grown in nutrient broth for investigating the growth of S. Enteritidis S003” (line174-176). How does the S. Enteritidis S003 grow in ordinary NB medium?

5. The first paragraph of Discussion describes the research progress of probiotics in bacteriostasis. These contents should be moved into the Introduction part.

---

## Round 0.2 · accepted · Accept

Dear Author, I am glad to inform you that your manuscript has been accepted for publication in PeerJ journal.